# Long-term time-lapse microscopy of *C. elegans* post-embryonic development

Nicola Gritti[1], Simone Kienle[1], Olga Filina[1] & Jeroen Sebastiaan van Zon[1]

We present a microscopy technique that enables long-term time-lapse microscopy at single-cell resolution in moving and feeding *Caenorhabditis elegans* larvae. Time-lapse microscopy of *C. elegans* post-embryonic development is challenging, as larvae are highly motile. Moreover, immobilization generally leads to rapid developmental arrest. Instead, we confine larval movement to microchambers that contain bacteria as food, and use fast image acquisition and image analysis to follow the dynamics of cells inside individual larvae, as they move within each microchamber. This allows us to perform fluorescence microscopy of 10–20 animals in parallel with 20 min time resolution. We demonstrate the power of our approach by analysing the dynamics of cell division, cell migration and gene expression over the full ~48 h of development from larva to adult. Our approach now makes it possible to study the behaviour of individual cells inside the body of a feeding and growing animal.

[1] FOM Institute AMOLF, Science Park 104, Amsterdam 1098 XG, The Netherlands. Correspondence and requests for materials should be addressed to J.S.v.Z. (email: j.v.zon@amolf.nl).

Recent advances in microscopy have made it possible to follow the dynamics of many, if not all cells in the development of entire zebrafish and fruit fly embryos[1]. However, in these model organisms time-lapse microscopy is typically restricted to early stages of embryonic development. Owing to their small and transparent anatomy, nematodes such as *Caenorhabditis elegans* are currently the only animals in which the entire development from embryo to adult can in principle be studied with single-cell resolution[2–5]. This also makes *C. elegans* uniquely suited to study the interplay between development and environmental cues such as diet, food availability and pheromones[6–8].

However, long-term time-lapse microscopy is currently rarely used to study *C. elegans* post-embryonic development. This is because *C. elegans* larvae are highly motile and thus are difficult to image at high magnification. Immobilizing larvae either mechanically or by paralysis-inducing drugs allows time-lapse microscopy only for limited time periods, as it prevents the animal from feeding, resulting in developmental arrest within hours[9,10]. Microfluidics has been used to immobilize nematodes for microscopy by mechanical clamping[11,12], flow[13,14] or changes in the physicochemical environment[15–17]; however, most of these devices are geared towards immobilizing adult nematodes and are not designed to support sustained development. Experiments that did support normal larval growth so far lacked the resolution to study development at the single-cell level[18–20].

To perform time-lapse microscopy of *C. elegans* post-embryonic development we instead use a different approach (Fig. 1): first, we constrain larval movement to the field of view of the microscope using microfabricated hydrogel chambers containing bacteria as food. Next, we use fast image acquisition to capture sharp images of larvae as they move inside each microchamber, precluding the need for immobilization altogether. Finally, we use image analysis to track the dynamics of cells inside the animal's body. Microchambers have two main advantages over active microfluidics: first, they are simple to use, requiring no moving parts or flow. Second, in contrast to microfluidics, microchambers do not require using liquid culture. Instead, animals move and feed under conditions similar to standard *C. elegans* culture on agar plates and the established microscopy protocols for studying nematode development[2]. Hydrogel microchambers have been used to constrain nematode movement for studying behaviour[21], but so far not development.

Here we show that, using arrays of microchambers, we can perform fluorescence microscopy of developmental dynamics in 10–20 animals simultaneously, with 20 min time resolution for the full ~48 h of post-embryonic development. To demonstrate the power of our approach we measured, in single animals, the dynamics of (i) seam cell divisions, (ii) distal tip cell (DTC) migration and (iii) molting cycle gene expression oscillations—three processes that because of their ~30–40 h duration were so far inaccessible for immobilization-based time-lapse microscopy. The control of cell division, cell migration and gene expression is the hallmark of development, and our analysis shows that the dynamical information captured by our approach can provide new insight into the mechanisms that control these processes. In general, we expect that the ability to follow individual cells in freely moving and growing animals will provide an unprecedented view on development.

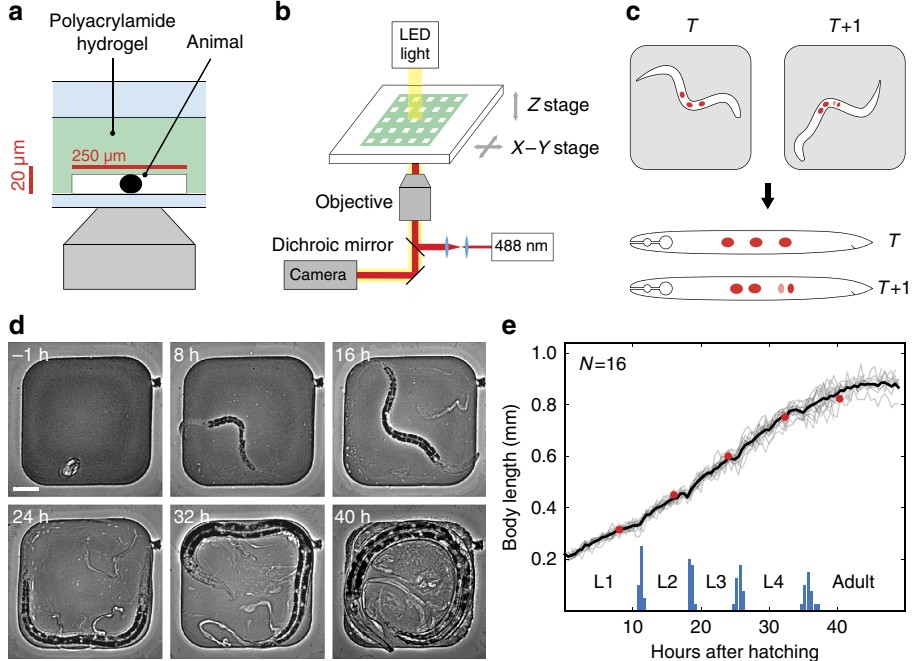

**Figure 1 | Imaging development of nematodes in polyacrylamide microchambers.** (**a**) Schematic cross-section of a single microchamber. The hydrogel layer is clamped between a coverslip (blue, bottom) and a cover glass (blue, top). (**b**) Imaging set-up. To image animals moving within microchambers, we used LED and laser illumination to achieve short (1–10 ms) exposure times and a fast piezo Z-stage to move rapidly between imaging planes. To image multiple animals in parallel, an X–Y motorized-stage cycled between individual microchambers in a microchamber array (green). (**c**) Image analysis. For each image, the body axis and positions of fluorescently labelled cells (red) are manually annotated. Cell positions are then converted to body axis coordinates to allow systematic comparison between time points. Finally, cell divisions, cell movements or changes in gene expression are recorded. (**d**) Images of a single growing animal in a 250 μm × 250 μm × 20 μm microchamber. Time is indicated in hours after hatching. Old cuticles, which are shed at the end of each larval stage (L1–L4), are also visible. Scale bar, 50 μm. (**e**) Body length (grey lines for individual animals and black line for population average) and fraction of animals in ecdysis (blue bars) as a function of time for N = 16 animals. Red markers correspond to the body length of the animal shown in **d**. Time of ecdysis is defined by the appearance of a newly shed cuticle in the microchamber.

## Results

**Larval development in microchambers.** To constrain *C. elegans* larvae to the field of view of the microscope, we microfabricated 250 µm × 250 µm × 20 µm chambers in a 10% polyacrylamide hydrogel (Fig. 1a). We created 10 × 10 microchamber arrays from a master mold created with standard soft-lithography techniques (Methods). To fabricate chambers we used polyacrylamide rather than agarose hydrogels, as used previously[21], because in our hands thin polyacrylamide layers were less brittle and easier to handle[22]. We filled the chambers with a single *C. elegans* egg and *Escherichia coli* OP50 bacteria as food (Methods). Subsequently, we clamped the polyacrylamide microchamber array between a standard microscope slide and a glass coverslip to prevent the sample from drying during the experiment.

Upon hatching, larvae moved and fed inside the microchambers (Fig. 1d and Supplementary Video 1). Larval movement can be fast (peaking at $50 \, \mu m \, s^{-1}$ for reversals). To minimize larval movement during image acquisition, we optimized our microscopy set-up for short acquisition times (Fig. 1b). First, we used light-emitting diode (LED) trans-illumination and laser epi-illumination to reduce exposure times to 1–10 ms. Second, we used a fast piezo Z-stage to scan the microchamber in the axial direction. Together, this enables us to acquire Z-stacks of 20–30 Z-slices with two imaging channels in <500 ms. By combining a sCMOS camera with a large camera chip (2,048 × 2,048 pixel, 6.5 µm × 6.5 µm per pixel) with a high numerical aperture (NA) 40× objective, we could image the entire microchamber (∼250 µm) while still resolving subcellular features (∼0.3 µm for 1.3 NA). By moving between individual chambers in the microchamber array using an X–Y motorized stage, we routinely imaged 10–20 larvae in a single imaging session.

We observed that after hatching individual larvae developed into adult animals over the course of ∼40 h, without leaving the chamber (Fig. 1d). To confirm that the observed growth corresponded to normal development, we measured two markers of developmental progression. *C. elegans* development is divided into four larval stages, labelled L1–L4. Animals molt at the end of each larval stage, an event called ecdysis. We first measured, for each animal, the time of all ecdyses, marked by the appearance of an old cuticle inside the chamber (Fig. 1d). The observed duration of each larval stage (average and standard deviation (s.d.) are 11.1 ± 0.2, 7.3 ± 0.2, 7.1 ± 0.3 and 10.2 ± 0.4 h for L1–L4) agreed with established values under standard culturing conditions[23]. Second, we measured body-length extension as a function of time (Fig. 1e). We found that body length varied between individual time points, likely reflecting the deformability of the animal's body. However, on average, we observed that body length increased with a fixed, larval stage-dependent rate, with pauses in growth observed before molts, as observed previously[20,23]. Body-length extension in microchambers agreed well with growth as observed on standard agar plates and occurred with limited compression of the animal in the vertical direction (Supplementary Fig. 1). Moreover, the body length at the start of each larval stage agreed well with previous measurements[23]. Together, this showed that *C. elegans* larvae developed normally inside our microchambers.

We found significant animal-to-animal variability, both in timing of ecdysis and body-length extension (Fig. 1e), even in animals imaged simultaneously. Similar variability was observed recently in *C. elegans* larvae developing in liquid culture[20]. However, to exclude that this variability was because of insufficient food in the microchambers, we also quantified timing of ecdysis and body-length extension in animals contained in larger microchambers (290 µm × 290 µm × 25 µm; Supplementary Fig. 1a). We observed no changes in the dynamics of development nor a decrease in animal-to-animal variability.

This suggested that the observed variability is intrinsic to *C. elegans* development.

**Seam cell lineage.** Because of its invariant cell lineage, *C. elegans* is uniquely suited to study the genetic control of cell lineages[2,24]. However, obtaining lineages remains laborious as it relies on manual observation over extended time periods. To test whether our set-up could simplify lineage analysis, we used it to study the seam cell lineage. Seam cells form a row of cells along the left and right sides of *C. elegans* animals (Fig. 2a) that divide with a complex pattern of asymmetric and symmetric cell divisions over an ∼40 h period[2]. Asymmetric divisions result in a new undifferentiated seam cell and a differentiated hypodermal (H1, H2, V1–V6, T) or neuronal/glial cell (H2, V5, T). In the L2 stage, this is preceded by a symmetric division that doubles the seam cell number (H1, V1–V4, V6). At the L4 molt, the remaining seam cells terminally differentiate. Because of the long duration, full seam cell lineages have never been imaged in a single animal. As a consequence, it remains poorly understood how the timing of seam cell divisions is controlled.

To visualize the seam cells, we used a strain, *wIs51[SCMp::GFP]*, that carries a nuclear seam cell marker[25]. This marker is sufficiently bright that we could visualize seam cells on both sides of the body over all four larval stages (Fig. 2c and Supplementary Video 2). We detected cell divisions by the first appearance of two daughter nuclei at the position previously occupied by the mother cell. We could unambiguously assign the fate of each daughter cell, as only the seam cell daughter retained nuclear fluorescence. In this way, we could reconstruct the full lineage for all seam cells (Fig. 2b).

Our analysis extends the standard lineaging approach[2] by providing the exact time of each division relative to the time of hatching, allowing us to study variation in timing both between seam cell lineages and within each lineage between animals. We first compared the average cell division timing between the different lineages (Fig. 2d). We measured, at each larval stage, the division time of each individual seam cell with respect to the average division time of all seam cells in the same animal. We found that seam cell divisions followed a particular sequence, with seam cells at the centre of the body (V2–V4) on average dividing before those closer to the head and tail (H1, H2, V6, T). This difference in timing is most pronounced in the earlier larval stages. The main deviation from this sequence was V5, which typically divided first in the L1 and L2 larval stages. However, we observed significant variability around the mean division times (Fig. 2d), leading to deviations from the average sequence. For instance, in 3/16 animals other seam cells divided before V5 at the L1 stage (Supplementary Fig. 2). We also observed significant animal-to-animal variability in division time within seam cell lineages (Fig. 2e), with typical s.d. of ∼0.3 h. Such quantitative measurements of average timing and variability can contribute towards understanding the cues that trigger seam cell divisions.

Next, we tested whether our set-up could aid the analysis of lineage mutants. We focused on an uncharacterized mutant strain, MBA48 (a gift from M. Barkoulas, Imperial College London), that exhibited variable seam cell numbers in adult animals, as observed using the *wIs51* seam cell marker. Variable lineage mutants are difficult to study as they require obtaining lineages in multiple animals. To find the origin of the variability in seam cell number in MBA48, we determined full seam cell lineages for multiple animals. We observed two deviations from the wild-type lineage: (i) conversion of asymmetric divisions into symmetric divisions that yielded two seam cells (62/456 divisions, Fig. 2f and Supplementary Video 3) and (ii) seam cells failing to

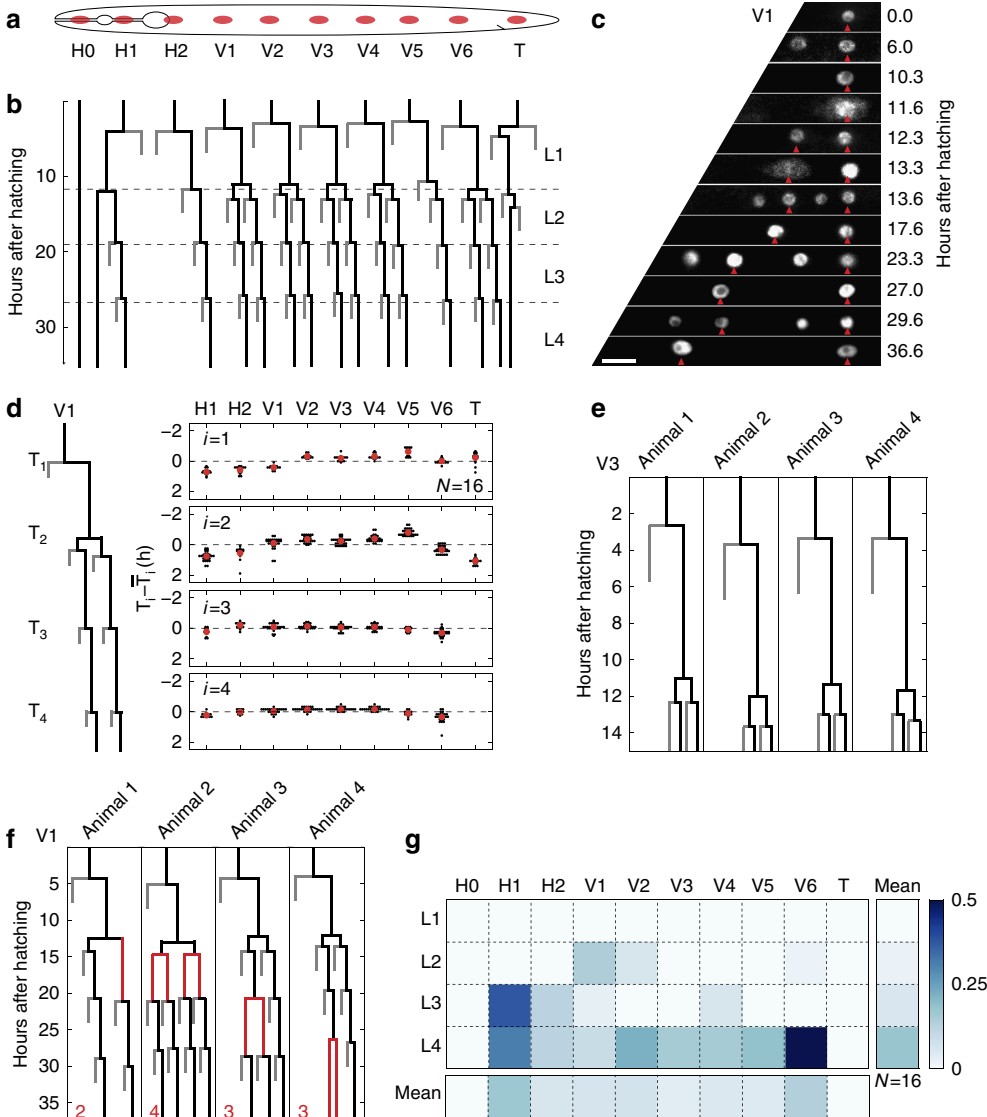

**Figure 2 | Seam cell lineage. (a)** Position of seam cells (in red) along the body axis. **(b)** Example of seam cell lineages measured in a single animal. Black lines represent seam cells and grey lines differentiated cells. Dashed lines indicate time of ecdysis, separating the different larval stages, L1–L4. Divisions are indicated at the exact time of occurrence, with 20 min resolution. **(c)** Image sequence of the V1 lineage in a single animal carrying a *wIs51[SCMp::GFP]* nuclear seam cell marker. Seam cell nuclei are indicated by red arrows. Other nuclei belong to hypodermal cells. Images were computationally straightened and aligned to the posterior-most seam cell. Scale bar, 10 µm. **(d)** Analysis of cell division timing. For each seam cell division *i*, we plot the relative division time $T_i - \bar{T}_i$ (black markers), where $T_i$ is the cell division time and $\bar{T}_i$ is the division time averaged over all nine lineages, H1-T, on both sides of the animal. Also shown is the relative division time averaged over all animals (red markers). **(e)** Animal-to-animal variability in cell division time in the first three divisions of the V3 lineage. **(f)** Examples of V1 lineages in different MBA48-mutant animals. Red lines represent lineage errors. The final number of seam cells is indicated for each lineage. **(g)** Occurrence of lineage errors. For each lineage and larval stage, colour represents the probability of errors. Also shown are the mean probability of errors for each larval stage (right-most column) and each lineage (bottom row).

divide (9/456 divisions). Note that in the latter case the number of hypodermal cells is reduced, but the final seam cell number is unaffected. Hence, such deviations cannot be identified when seam cells are counted only at a single developmental stage. Both types of deviations occurred stochastically, predominantly in the L3 and L4 stages and were distributed unequally over the different lineages, most strongly impacting the H1 and V6 lineages (Fig. 2g). Hence, our results show that stage- and lineage-specific differences exist in the regulation of seam cell divisions.

**Cell migration.** Cell migration is an essential aspect of development and, given its highly dynamical nature, time-lapse microscopy is a powerful tool to study it. Q neuroblast migration has been imaged using time-lapse microscopy for short (<3–4 h) periods in immobilized *C. elegans* larvae[9]. However, so far, no post-embryonic cell migration in *C. elegans* has been visualized in full. To test our set-up, we therefore attempted to image DTC migration, which is the longest cell migration process in *C. elegans* development[26]. The two DTCs are born at the L1 molt and guide the shape of the gonad by migrating along a stereotypical path over an ~30 h period (Fig. 3a): during the L2 and L3 stages the DTCs migrate outwards, one moving anteriorly and the other posteriorly. In the late-L3 stage, both DTCs turn and move from the ventral to the dorsal sides of the body. Finally, in the L4 stage the DTCs move back to the centre of the body.

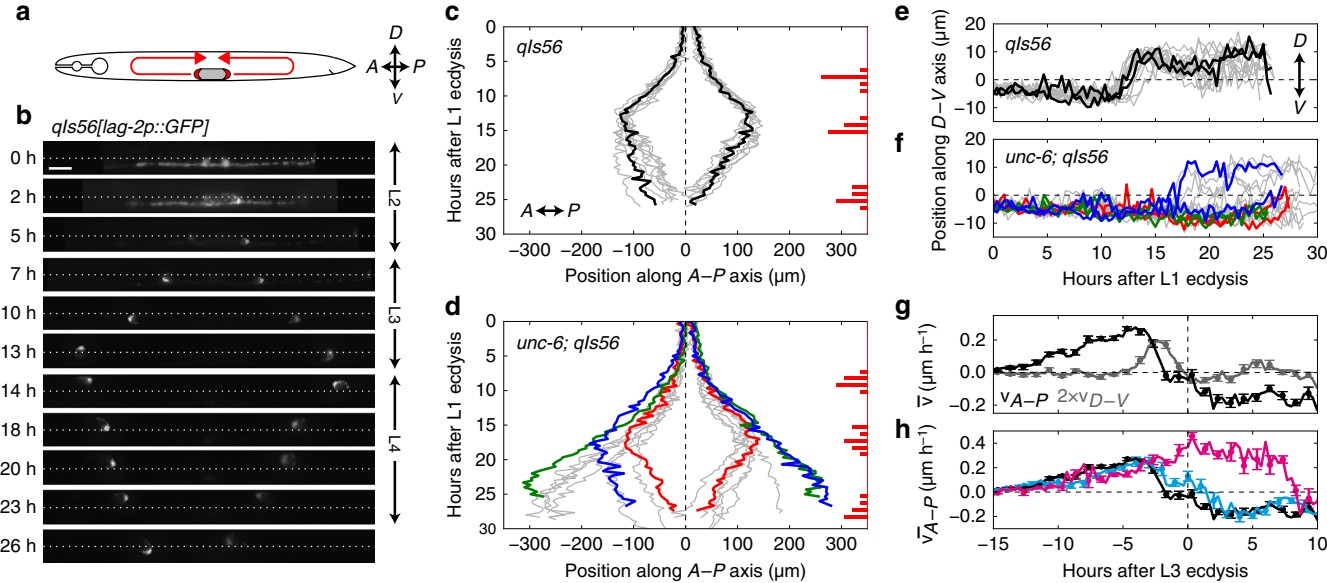

**Figure 3 | DTC migration.** (**a**) Overview showing the gonad (grey) and the DTC (red) migration path. (**b**) Computationally straightened microscopy images of a single animal carrying the *qIs56[lag-2p::GFP]* DTC body marker at different times after the L1 ecdysis. Dotted line indicates the central body axis. Orientation of A–P and D–V axes is as in **a**. Scale bar, 20 μm. (**c**) A–P position as function of time in *qIs56* animals ($N = 10$). Dashed line corresponds to the midbody. DTC trajectories for the animal in **b** are shown in black. Red bars show the fraction of animals in ecdysis as a function of time. (**d**) Same as **c** but for *unc-6(ev400);qIs56* mutants ($N = 9$). Coloured lines indicate animals in which DTCs moved inwards ventrally (red), DTCs failed to turn inwards (green) or a single DTC migrated dorsally (blue). (**e**) D–V position as a function of time. Dashed line corresponds to the central body axis. Trajectories for the animal in **b** are shown in black. (**f**) Same as **e**, but for *unc-6;qIs56* mutants. Coloured lines correspond to the animals highlighted in **d**. (**g**) Average migration velocity $\bar{v}$ in wild-type animals as a function of time after L3 ecdysis. $v_{A-P} > 0$ indicates outward movement and $v_{A-P} < 0$ inward movement. Error bars are s.e.m. (**h**) Average A–P velocity for DTCs in wild-type animals (black), DTCs in *unc-6;qIs56* mutants that moved inwards after the L3 ecdysis (cyan) and anterior DTCs in *unc-6;qIs56* mutants that only moved outwards (magenta).

DTC migration is an important model system for understanding the genetic regulation of complex migratory trajectories[27]. However, this is mostly studied by observing DTC positions at a single developmental stage and information about dynamics is very limited. In particular, it is unclear how kinetic parameters such as speed and direction are controlled to produce the correct migration path.

We visualized DTC migration in *qIs56[lag-2p::GFP]* animals, where the DTC bodies are fluorescently labelled[28]. We could follow the full DTC migration in individual animals (Fig. 3b and Supplementary Video 4). To compare the DTC trajectories between animals, we quantified both the positions along the anteroposterior (A–P) axis (Fig. 3c) and the dorsal–ventral (D–V) axis (Fig. 3e). We found that the dynamics of outward migration in the L2 and L3 stages was highly reproducible, but observed increased variability during the inward migration in the L4 stage, which was more pronounced in the anterior DTC (s.d. in position $\sigma_{A-P} = 27$ μm at the L4 ecdysis) compared with the posterior DTC ($\sigma_{A-P} = 12$ μm; Fig. 3c). The movement from the ventral to the dorsal sides was rapid, occurring in ~3 h (Fig. 3e). On average, we found that the anterior DTC crossed the midline of the body first, although we observed significant variability in the difference in crossing times between the two DTCs (Supplementary Fig. 3).

D–V migration occurred at the end of the L3 stage and was accompanied by a slowdown of A–P migration (Fig. 3c). To study the coordination between A–P and D–V migration in more detail, we measured the average migration velocity $\bar{v}_{A-P}$ and $\bar{v}_{D-V}$ as a function of time, using the L3 ecdysis as a reference point (Fig. 3g). We found that changes in migration dynamics were tightly coordinated: A–P migration ceased and D–V migration started simultaneously ~3h before the L3 ecdysis

and resumption of A–P migration occurred exactly after the L3 ecdysis. The latter observation is particularly striking as the exact time of L3 ecdysis varied significantly between animals (Fig. 3c, red bars). Previous studies already indicated that DTC migration is controlled by developmental timing cues[29,30]; however, our analysis of the kinetics of DTC migration provides strong evidence for a tight link specifically between the turning events and the molting cycle.

To test whether our measurements of DTC migration dynamics could also give insight into the mechanisms that control the migration direction, we followed DTC migration in an *unc-6(ev400); qIs56* mutant[31]. D–V migration in DTCs is controlled by Netrin signalling, with the ligand UNC-6/Netrin proposed to form a D–V gradient[32]. In *unc-6* mutants, DTCs often fail to migrate dorsally[31]. We observed that DTC migration in *unc-6;qIs56* mutants was highly variable, with 11/18 DTCs migrating exclusively on the ventral side (Fig. 3d,f and Supplementary Video 5). Independent of errors in D–V migration, DTCs also failed to turn inwards and continued towards the head or tail (10/18 DTCs). D–V migration of DTCs is thought to be controlled by modulating their sensitivity to UNC-6 at the appropriate time[30]. However, it is unknown how the cessation of A–P movement during D–V migration is controlled. To examine this we measured the average migration velocity $\bar{v}_{A-P}$ in *unc-6;qIs56* mutants (Fig. 3h). We found that those DTCs that turned inwards at the L3 ecdysis stopped A–P movement, similar to wild-type animals. However, anterior DTCs that failed to turn inwards showed no reduction in A–P velocity. This result suggested that for anterior DTCs there is no temporal cue that specifically inhibited A–P movement during D–V migration. Rather, the decrease in A–P movement seemed linked to the reversal in direction of A–P migration. In contrast, in the small

number of posterior DTCs that only migrate outwards, A–P movement did appear to cease at the normal time of D–V migration (Supplementary Fig. 3), suggesting differences in the control of migration between the two DTCs.

**Oscillatory gene expression**. So far, we used fluorescence only to determine cell position. We next tested whether we could also quantify fluorescence intensity as a measure of gene expression dynamics, focusing on the oscillatory expression of molting cycle genes. Dynamic regulation of gene expression is essential for development. A striking example is provided by the molting cycle genes in *C. elegans*, whose expression peaks once every larval stage[33]. Moreover, recent RNA-sequencing experiments found that many genes exhibited oscillatory expression in phase with the molting cycle[34,35]. However, so far, such gene expression oscillations were characterized at the population level, but not in single animals or single cells. Hence, it is not known with what precision the period of the oscillations is controlled and how strongly they are synchronized within the body.

First, we characterized the expression dynamics of the molting cycle gene *mlt-10*, which is essential for molting and expressed in the hypodermis only during the molt. We studied a transcriptional reporter strain, *mgIs49[mlt-10p::GFP-PEST]*, used previously to characterize *mlt-10* expression dynamics at the population level[33]. We could follow expression dynamics in the *mlt-10* reporter for all four larval stages, with a clear pulse in fluorescence intensity observed close to each ecdysis (Supplementary Video 6). To study the spatiotemporal *mlt-10* expression dynamics, we quantified both the average fluorescence intensity along the A–P axis (Fig. 4a,b) and the total fluorescence intensity (Fig. 4c) as a function of time. We found that the oscillatory dynamics was uniform, that is, with a phase independent of the A–P position (Fig. 4a,b). We observed significant animal-to-animal variability in the exact timing of the *mlt-10* expression peak (Fig. 4d, $28 \pm 1$ h for the L3 peak). However, *mlt-10* expression dynamics was tightly correlated with the subsequent ecdysis (Fig. 4e, peaking $1.1 \pm 0.1$ h before L3 ecdysis).

The *mlt-10* reporter was expressed in many cells. To test whether we could follow expression dynamics with single-cell resolution, we also measured *wrt-2* expression dynamics. The gene *wrt-2* is expressed exclusively in seam cells[36] and was recently found to exhibit oscillatory gene expression at the population level in the L3–L4 stage[35]. We followed *wrt-2* expression in *heIs63[wrt-2p::H2B::GFP, wrt-2p::PH::GFP]* animals, in which green fluorescent protein (GFP) is targeted both to the seam cell nucleus and membrane[37]. We found that the fluorescence signal was bright enough to visualize expression in seam cells on the side of the animal closest to the objective, but not on the opposite side, likely because of light scattering in the intervening tissue (Supplementary Fig. 4). As animals sometimes flip from one side to the other around the molt[38], we could not follow single seam cells over the entire course of development. However, focusing only on the seam cells closest to the objective, we could clearly observe oscillations in *wrt-2* expression in single seam cells over all four larval stages (Fig. 4f and Supplementary Video 7). To quantify *wrt-2* expression, we measured the total nuclear fluorescence intensity in the seam cells V1–V5 as a function of time (Fig. 4g). We found that both the period and phase of *wrt-2* oscillations agreed with previous measurements of *wrt-2* mRNA dynamics[35]. Moreover, *wrt-2* oscillations were strongly correlated even between seam cells such as V1 and V5 that reside in different parts of the body (Fig. 4h). Similar to *mlt-10* expression oscillations, we observed that, while there existed significant animal-to-animal variability in the exact time

of the *wrt-2* expression peaks (Fig. 4d,i, $26 \pm 1$ h for the L3 peak), the expression peaks were nevertheless precisely timed with respect to the ecdysis (Fig. 4e, peaking $1.5 \pm 0.4$ h before L3 ecdysis). In general, for both *mlt-10* and *wrt-2* we find that, despite clear animal-to-animal variability, the timing of expression peaks was tightly coupled to ecdysis and was strongly correlated between cells at different positions. This suggests that molting cycle gene expression and ecdysis are under strong global control.

## Discussion

Here we describe a technique to perform time-lapse microscopy in moving *C. elegans* larvae, with single-cell resolution and for the full duration of post-embryonic development. We achieved this by using hydrogel microchambers that confine animal movement to the field of view of the microscope while containing sufficient food for development. Owing to our use of high NA objectives, our approach is comparable in spatial resolution to existing immobilization-based time-lapse microscopy techniques, although the requirement for short exposure times to image moving animals makes it more challenging to combine with confocal microscopy to increase the axial resolution. We can envision several ways in which to expand on the design of our set-up. First, it could be combined naturally with RNA interference by filling microchambers with bacteria that express the desired double stranded RNA[39]. Similarly, the effect of diet on development could be studied at the single-cell level by selecting different bacterial strains as food source[40]. Finally, a separate microfluidic layer on top of the hydrogel chambers could be used to change the local chemical environment[22] in <10 min to deliver environmental cues, for example, dauer pheromone, at precisely timed developmental stages.

We used our technique to study three processes that, because of their long duration (30–40 h), had been inaccessible for time-lapse imaging: seam cell divisions, DTC migration and molting cycle gene expression oscillations. We were able to perform detailed analysis of the timing, kinetics and variability in these processes, both in wild-type animals and mutants, an approach that can be extended to many other developmental processes in *C. elegans*. We typically imaged 10–20 animals per imaging session, even though, in principle, the combination of short (<1 s per animal) acquisition time and 20 min interval between acquisitions allows for hundreds of animals to be imaged simultaneously. This is because the bottleneck was formed by the manual annotation of the animal's body axis and cells. It should be possible to optimize this by improved automation and image analysis. We believe that our set-up could make long-term time-lapse microscopy a routine tool to study *C. elegans* post-embryonic development, with the potential to significantly increase our understanding of lineage control, morphogenesis and regulation of gene expression.

## Methods

***C. elegans* culture and strains**. All *C. elegans* strains were cultured at 20 °C on Nematode growth medium (NGM) agar plates with OP50 bacteria. Wild-type nematodes were strain N2. The following mutations and integrated transgenic arrays were used: LGIV: *mgIs49 [mlt-10p::GFP-PEST]*[33], LGV: *wIs51 [SCMp::GFP]*[25], *qIs56 [lag-2p::GFP]*[28], *heIs63 [wrt-2p::GFP::PH; wrt-2p::GFP::H2B]*[37] and LGX: *unc-6 (ev400)*[31]. The mutant strain MBA48 was a gift from Michalis Barkoulas, Imperial College London, and will be described in detail elsewhere.

**Microchamber fabrication**. Microfabricated arrays of chambers were made from a master mold as described in ref. 22. Master molds were created using standard soft-lithography techniques. Briefly, SU-8 2025 epoxy resin (MicroChem) was first spin-coated on a silicon wafer to form a 20 μm layer. The SU-8 layer was exposed with ultraviolet-light through a foil mask (SELBA S.A.) containing the micropattern (Supplementary Fig. 5). Microchamber dimensions are 250 × 250 × 20 μm for all experiments, unless specified otherwise. To create poly-acrylamide microchamber arrays from the master mold, a 10% dilution of 29:1

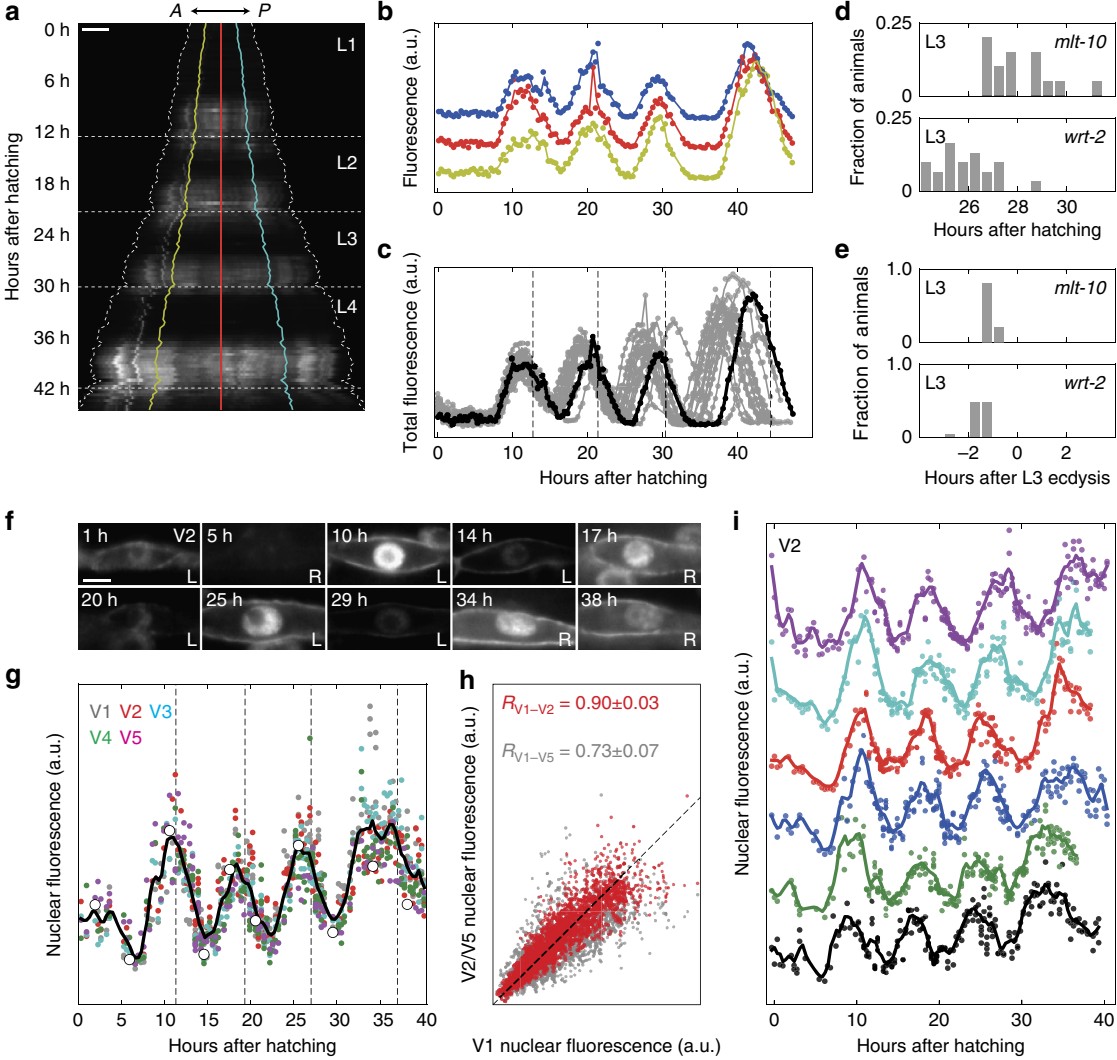

**Figure 4 | Oscillatory gene expression.** (**a**) Kymograph of *mlt-10* expression along the A–P axis as a function of time in a single *mgIs49[mlt-10p::GFP-PEST]* animal. Dotted lines represent the position of head and tail, and horizontal dashed lines represent ecdysis. Coloured lines indicate the regions evaluated in **b**. Scale bar, 100 µm. (**b**) *mlt-10* expression oscillations at different A–P positions for the animal in **a**. (**c**) *mlt-10* expression integrated over the entire animal as a function of time for $N = 15$ animals. The *mlt-10* expression dynamics (black line) and time of ecdysis (dashed lines) are indicated for the animal in **a**. (**d**,**e**) Time distribution of the L3 peak in *mlt-10* ($N = 15$) and *wrt-2* ($N = 23$) expression relative to (**d**) time of hatching and (**e**) time of L3 ecdysis. (**f**) *wrt-2* expression oscillations in the posterior-most V2 seam cell in a *hels63[wrt-2p::H2B::GFP, wrt-2p::PH::GFP]* animal. Time is hours after hatching. The label indicates whether the cell is the left (L) or right (R) V2 cell. Scale bar, 5 µm. (**g**) Single animal *wrt-2* expression oscillations. White markers correspond to the images in **f**. The black line represents a sliding average with 2 h window size over V1–V5. (**h**) Correlation in *wrt-2* expression between the V1 and V2 (red) and V5 (grey) cells. Markers and correlation coefficient $R$ are for $N = 23$ animals over all larval stages. (**i**) *wrt-2* expression oscillations in the V2 seam cell lineage in different animals. Lines correspond to sliding average with 2 h window size.

acrylamide/bis-acrylamide was mixed with 0.1% ammonium persulfate (Sigma) and 0.01% TEMED (Sigma) as polymerization initiators. The resulting aqueous solution was then poured in a cavity placed on top of the micropatterned silicon wafer. The cavity was closed with a silanized coverslip and sealed by mechanical clamping, allowing the solution to polymerize for 2 h. To remove the toxic unpolymerized acrylamide monomers, the resulting polyacrylamide microchamber arrays were washed at least three times for at least 3 h each in distilled water. Fewer or shorter washing steps often resulted in developmental arrest. Microchamber arrays could be stored in distilled water for ~15 days. Single microchamber arrays were placed in M9 buffer for 4 h directly before time-lapse imaging.

**Sample preparation.** To prepare the sample, a glass spacer with the same height as the polyacrylamide membrane was attached to a 76 × 26 × 1 mm microscope slide using high vacuum grease (Dow Corning). A single microchamber array was positioned with tweezers on the microscope slide, with the openings of the microchambers facing up. Excess liquid was removed with a tissue. With a pipette, drops of M9 buffer (~40 µl in total) were placed on the side and on the surface of the microchamber array, while preventing the liquid from filling the chambers. To load *C. elegans* embryos into the microchambers we followed the approach in

ref. 21. Under a dissection microscope, a drop of bacterial suspension containing a single late-stage embryo was transferred from a NGM agar plate seeded with OP50 bacteria into a microchamber, using an eyelash attached to a Pasteur pipette. To facilitate the release of the bacteria and embryo into the chamber, the eyelash was dipped briefly into the M9 drop before touching the microchamber. Once the egg was transferred, more bacterial suspension was added to the microchamber using the eyelash, until completely filled. For each experiment, ~10–20 chambers were loaded. Subsequently, tissue paper was used to remove all excess M9 buffer. Finally, a 25 × 75 mm #1 coverslip was lowered on the chambers to seal the sample, slow enough to avoid forming large air bubbles. The sample was placed on a custom fabricated holder and clamped to seal the chambers and avoid liquid evaporation during the duration of the experiment (Supplementary Fig. 5).

**Microscopy imaging.** We performed time-lapse imaging on a Nikon Ti-E inverted microscope. Using a large chip camera (Hamamatsu sCMOS Orca v2), it was possible to fit single microchambers in the field of view of the camera while using a 40 × magnification objective (Nikon CFI Plan Fluor 40 ×, NA = 1.3, oil immersion). Transmission imaging was performed using a red LED (CoolLED pE-100 615 nm), while fluorescence images were acquired using a 488 nm laser (Coherent

OBIS LS 488–100). The laser beam was expanded from 0.7 to 36 mm through a telescope composed of two achromatic lenses of 10 and 500 mm focal lengths (Thorlabs AC080-010-A-ML and AC508-500-A). The expanded beam was then aligned through additional dielectric mirrors (Thorlab BB2-E02) to enter the back aperture of the microscope. A tube lens (300 mm focal length, Thorlabs AC508-300-A) was used to focus the beam in the back focal plane of the objective. For fluorescence microscopy, we used a dual band filter set (Chroma, 59904-ET). An $XY$-motorized stage (Micro Drive, Mad City Labs) was used to move between chambers, while a piezo Z-stage (Nano Drive 85, Mad City Labs) was used to move the sample in the Z direction. To optimize acquisition speed, we synchronized the camera, laser illumination and stage movement as follows: to operate the rolling-shutter camera in the global exposure mode, the laser beam was switched on (rise time $<3\,\mu s$) once all the lines on the camera chip were active and switched off once the camera started reading out the chip. In order to rapidly acquire Z-stacks, we synchronized the piezo Z-stage and the camera, so that the stage moved to the new Z position during the 10 ms that the camera read out the chip to its internal memory. The microscope and all its components were controlled with custom software implemented using a National Instruments card (PCIe-6323) installed on a computer with a solid-state drive (Kingston V300-120GB), an Intel Core i7 processor (3.50 GHz) and 16 GB of RAM. By using sufficiently high laser power (80–100 mW), we could use exposure times that were short enough (1–10 ms) that animal movement during acquisition was negligible. Acquiring a single imaging volume, typically consisting of 20 Z-slices in two channels, took $<0.4$ s. Some animal movements along the A–P axis was observed between Z-slices, particularly in L3–L4 larvae directly after the molt. Each chamber was imaged every 20 min for $\sim 48$ h. We found that shorter time intervals sometimes led to larvae arresting in the L1 larval stage. During imaging intervals, we used the Perfect Focus system of the microscope to prevent sample drift. Images were acquired in a temperature-controlled room at 22 °C.

**Image analysis.** Custom Python software was used to analyse the acquired images. For all experiments, we used the transmitted light images to record the time of hatching and ecdysis as well as the body length and position of the gonad (L2–L3) or vulva (L3–L4). We obtained the body axis by manually selecting 10–20 points on the animal's centre line and subsequently fitting a spline curve $\bar{x}(s)$ to these points, with $s$ being the arc length of the spline curve along the A–P axis. The body length was then given by the length of $\bar{x}(s)$. We measured the position of the gonad or vulva, as markers of the ventral side of the body axis, to establish the left–right orientation of the animal's body. For all cells of interest, we manually obtained their position $\bar{r} = (x, y, z)$ in the coordinate system of the camera chip. To obtain the cell position in the body's coordinate system, we calculated $s$, the position along the A–P axis, and $t$, the position along the D–V axis as follows (Supplementary Fig. 6): the A–P position was given by the arc length $s$ that minimized the distance between $\bar{r}$ and $\bar{x}(s)$. Next, the magnitude of the D–V position $t$ was given by this minimal distance and the sign of $t$ was defined so that $t < 0$ for the ventral side. This coordinate transformation was also used to create the computationally straightened images of animals in Figs 2–4. All image and data analysis software are available at http://github.com/jvzonlab/timelapse-natcomm-2016.

**Seam cells.** The seam cells were identified by their position along the A–P axis. Initiation of cell division could be observed by the loss of nuclear fluorescence due to nuclear envelope breakdown. The division time was given by the first appearance of two smaller daughter nuclei at the old position of the mother cell. In the seam cell mutant MBA48, we defined a lineage error as the first point at which the (sub-)lineage deviated from the wild-type lineage. For instance, if an additional seam cell was erroneously generated, we did not score for errors in the sublineage produced by that seam cell. To achieve this, we calculated the probability $P(l,s)$ of a lineage error occurring in seam cell lineage $l$ at larval stage $s$ as follows. For each seam cell $i$, for example, V2L.pp, in animal $w$, we assign a division class $d_i = 0, 1, 2$ for no division, symmetric division and asymmetric division, respectively. We did so for all seam cells in wild-type (WT) and mutant (M) animals. The error probability is then given by $P(l,s) = \sum_w \sum_{i \in C_{l,s}} (1 - \delta_{d_i^{WT}, d_i^M}) / (\sum_w \sum_{i \in C_{l,s}} 1)$, where $\delta_{n,m}$ is the Kronecker delta and $C_{l,s} = C_{l,s}^{WT} \cap C_{l,s}^M$ is the list of seam cells that are present in both the wild-type lineage and the mutant animal $w$.

**Distal tip cells.** For both DTCs we calculated their position $(s,t)$ along the A–P and D–V axes. To correct for small movements between Z-slices, we measured the A–P displacement $\Delta s$ of anatomical markers such as the pharyngeal bulbs, vulva and anus between the two Z-slices that contained a DTC. We then corrected the A–P positions of the DTCs by subtracting the measured offset $\Delta s$. However, this correction for the animal's A–P movement resulted only in minor quantitative changes to the DTC position data. For the DTC analysis, $s = 0$ corresponded either to the midbody (L1–L3), defined as the A–P position exactly between the posterior pharyngeal bulb and the anus, or the A–P position of the invagination of the vulva (L3–L4), with these two measures coinciding in most animals. To calculate the DTC velocities $v_{A-P}$ and $v_{D-V}$, we first applied a sliding average with 1 h window size to the measured $(s,t)$ trajectories and then calculated velocities as the time derivative of the average $(s,t)$ trajectories (Supplementary Fig. 3).

**Molting cycle genes.** For *mlt-10* expression, the mean fluorescence intensity as function of A–P position $s$ was obtained by averaging the fluorescence over a D–V window of $|t| < 60\,\mu m$. The expression dynamics at different A–P positions was determined by integrating fluorescence intensity over a region of 5% of body length, centred at positions at 25, 50 and 75% of body length. To measure *wrt-2* expression in single seam cells, we manually labelled the nuclei of the V1–V5 seam cells on the side closest to the objective. As the size and shape of the nuclei changed over time, we used an image segmentation algorithm (Otsu's method) on a $5\,\mu m \times 5\,\mu m$ region around each nucleus to obtain a mask of the nucleus. We then computed the mean fluorescence intensity of the pixels within this mask. To detect the time of peaks in *mlt-10* and *wrt-2* gene expression, we applied a Gaussian filter with width of 1 h to each fluorescence intensity time series and obtained for each larval stage the time at which the averaged time series exhibited its maximum.

**Data availability.** The data that support the findings of this study are available from the corresponding author upon request. All analysis software are available at http://github.com/jvzonlab/timelapse-natcomm-2016.

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

## Acknowledgements

This research was supported by an ERC Starting Grant (338200-STOCHCELLFATE) and an NWO VIDI grant (680-47-529) to J.S.v.Z. We thank Michalis Barkoulas and Allison Frand for providing us with *C. elegans* strains. We also thank Sander Tans and Tom Shimizu for a critical reading of this manuscript.

## Author contributions

N.G. and J.S.v.Z. conceived the microchamber and microscope design. N.G. performed microfabrication and built the time-lapse microscopy system. N.G. and J.S.v.Z. designed all experiments. N.G. and S.K. performed all experiments. N.G. and J.S.v.Z. developed all image and data analysis software. N.G., S.K. and O.F. performed data analysis. N.G. and J.S.v.Z. wrote the article, with all authors reviewing the manuscript.

## Additional information

**Competing financial interests:** The authors declare no competing financial interests.

