## [Peer Review File · Nature Communications]

Reviewer #1 (Remarks to the Author)

The authors describe a simple, yet powerful imaging system (microchambers with 3D widefield microscopy) for monitoring the growth of nematode larvae, at the single cell level, over extended periods (30-40 hours). The authors convincingly demonstrate that they can follow the dynamics of seam cell divisions, distal tip cell migration, and oscillations in molting gene expression in individual animals - phenomena that have been otherwise difficult to study due to the need for moderate spatial resolution and high time resolution.

The ms is well written, the methods outlined in reasonable detail, and the approach likely immediately applicable and extensible to other labs due to its technical simplicity. I recommend publication after the following points have been addressed:

a Conspicuously missing is a methods section detailing the computer hardware and storage capability necessary to acquire the data. How large are the datasets for a typical imaging run on one animal? On more than one animal? What computer was used to acquire the data, and to process the data (provide specs)?

b The authors imply that the relatively small walls of the microchamber do not perturb development, yet it is obvious that by the end of the acquisition, the worm fills most of the microchamber volume. I would like to be more convinced that the confinement does not alter the underlying growth: the authors show in Supplementary Figure 1 that a very slightly larger chamber slightly affects the growth curve, and imply this is due to the lack of food. This point should be addressed (i.e., add more food to definitively test this hypothesis), and a significantly larger microchamber used to demonstrate that the walls do not significantly affect growth.

c related to this point: larva dimensions may have diameter > 20 μ m, while chamber "height" is 20 μ m. Are larvae compressed and what effects does this have on development?

d I assume the units of the y axis in Fig. 2d are in hours. Please make explicit in the figure.

e In figures 2e, 2f, it might be helpful to explicitly indicate 'animal 1', 'animal 2', etc. above each lineage.

f What is the y axis scale in Figures 4d, e?

g. I could not find temperature info for microchambers when imaging. Temperature affects developmental timing so this should be specified in methods if not already present.

h. analysis of Netrin mutations on DTC D-V migration: the number of cells with phenotypes (10 + 11) is greater than the number examined. Do some of the DTCs show multiple phenotypes? If so, this should be clarified, as in lines 187-190

Reviewer #2 (Remarks to the Author)

Gritti et al reported a microscopy method for long-term fluorescence time-lapse imaging of *C. elegans* larvae. They showed that they could use this protocol to study seam cell division, distal tip cell migration and gene expression. This technique can be useful in several cases such as the long-range distal tip cell migration. However, the following limitations preclude its publication in a journal such as Nature Communication.

1. The protocol is lack of sufficient spatial resolution for scientific discoveries. The authors used an X40 objective for imaging analysis; however, under such a setup, very limited subcellular structure can be visible. It is nowadays essential to study cellular processes during *C. elegans* development at the molecular or subcellular resolution. X40 objective does not enable the imaging of the cytoskeleton or polarity molecules or others. Seeing the cell morphology (e.g. distal tip cell,) or nuclei (seam cells) or global change of fluorescence (gene expression) will not advance our understanding of cellular mechanisms of these processes. They authors should demonstrate that X100 objective or even higher (e.g., X150) can fit in their protocol.

2. Similarly, the method does not offer the sufficient temporal resolution for discoveries. They authors claimed that their imaging time interval is 20 min, which is far too slow to document enough details. For example, the post-embryonic Q cell or P cell or seam cell division can be completed within 20 minutes. The previous study of Q cell asymmetric cell division used 20 second

time interval to follow the dynamics of centrosome and myosin during divisions and even used 5 sec time interval to visualize membrane dynamics during Q cell division, leading to an important discovery in asymmetric cell division (Ou et al. Science 2010). Moreover, the migrating cells always use less than 10 minutes to establish polarity and reorganize the actin cytoskeleton after division. Again, the protocol described in this manuscript will not be possibly to yield any information to address such problem.

3. The major issue of this manuscript is that the lack of the demonstration of the potential of yielding novel biological insights. Any fancy techniques will be quickly faded if there is no such potential regardless what the technique relies on.

Although the protocol may not be very valuable for cell biology, it may attract developmental biologists if (1) long-term (>10 hr) imaging is required and (2) the temporal/spatial resolution is not necessary. To this end, the manuscript will be very interesting to the readership in journals such as a Tool in Development.

Finally, the writing can be improved to be more scholarly.

1. Line 28-29, "However, fluorescence time-lapse microscopy is currently rarely used to study *C. elegans* post embryonic development." "rarely" may not be appropriate because a simple PubMed search revealed more than 20 primary research papers using live cell imaging of worm larvae.

2. Line 150: "Q neuroblast migration has been imaged using timelapse microscopy for short (~1h) periods in immobilized *C. elegans* larvae⁹". Q cell migration can be imaged for 3-4 hours (Ou & Vale, Journal of Cell Biology, 2009) without affecting the migration speed and distance compared with the original Sulston and Horvitz work (Dev Biol 1997). Q cell was only imaged for a couple of hours because its migration is completed within 3-4 hours. There is no point to image further. Incorrect citation/interpretation of the previous literature will not certainly help the novelty of the current protocol.

3. The authors can emphasize more that this protocol can be complementary to the existing protocols: Chai et al. reported high resolution imaging of L1 larvae whereas this protocol, as the author stated, will not work for the L1 larvae because the high power of lasers must be used. However, the protocol enables the long-term imaging of post-L1 processes that do not require high spatial and temporal resolutions.

Reviewer #3 (Remarks to the Author)

Gritti and coauthors describe a system for imaging the entirety of *C. elegans* postembryonic development in living worms at single-cell resolution. They demonstrate the value of the system by imaging and quantitative analysis of three processes that take too long to image continuously previously: distal tip cell migration, seam cell divisions, and gene expression that oscillates with molting cycles.

I view this as a technical tour de force that can have a strong practical impact on diverse areas of biology that depend on imaging a living organism. The writing is clear, and the data are beautiful and well presented both in images and videos and in quantitative analyses.

I consider this manuscript as a very strong candidate for publication in Nature Communications (or Nature Methods), with just one major concern: The exposure times used were so short, and the fluorescent markers presumably so highly expressed, that it will be difficult for others to make practical use of this method in many cases. This was no doubt needed because the worms were moving. For this reason, I urge the authors to attempt combining the new method with their methods in their reference 22, for flowing in drugs that can reversibly arrest movement, and show that this allows longer exposure times needed for more dimly fluorescent cells.

Minor concerns:

Lines 24-25: Other nematode species have been lineaged too.

Line 61: There should be some explanation of how only 20um tall is sufficient, since an adult worm

is much thicker.

Line 134: MBA48 sounds like it must be a strain name, not a mutant name.

Line 429: "center of the midbody" is redundant.

Reviewer #4 (Remarks to the Author)

The paper by Gritti et al describes a technique to perform long-term microscopy on developing and moving *C. elegans*. The paper provides three beautifully done examples of how this technique is used in developmental biology and fluorescence live microscopy.

While the work presented in the paper is executed extremely nicely, the technical advancement is not terribly major. The basic idea is to push the microscopy system to have a short exposure time such that the otherwise possibly blurred images would not be. The way the authors demonstrated is by using a strong excitation source, very sensitive camera, and very importantly, quite bright markers (as the authors remarked in the text, "sufficiently bright"). This is perhaps the key limitation of the technique - that the markers have to be sufficiently bright and that one has to have access to fairly expensive microscopy equipment.

The examples that the authors used are fairly well known developmental phenomena. While the authors observed that there exists an appreciable variability in each of the examples, and that the numbers of animals from these experiments are impressive, it doesn't seem that there is much novel biological insights from these experiments.

Taken together, the main complaint this review has is that there is not sufficient technical or scientific advancement, even though the work is very beautifully done and the paper is clearly and thoughtfully written.

Reviewer #1

The authors describe a simple, yet powerful imaging system (microchambers with 3D widefield microscopy) for monitoring the growth of nematode larvae, at the single cell level, over extended periods (30-40 hours). The authors convincingly demonstrate that they can follow the dynamics of seam cell divisions, distal tip cell migration, and oscillations in molting gene expression in individual animals - phenomena that have been otherwise difficult to study due to the need for moderate spatial resolution and high time resolution. The ms is well written, the methods outlined in reasonable detail, and the approach likely immediately applicable and extensible to other labs due to its technical simplicity. I recommend publication after the following points have been addressed:

We are happy to hear this positive assessment of the Reviewer.

a Conspicuously missing is a methods section detailing the computer hardware and storage capability necessary to acquire the data. How large are the datasets for a typical imaging run on one animal? On more than one animal? What computer was used to acquire the data, and to process the data (provide specs)?

We thank the Reviewer for pointing out this oversight. Briefly, we controlled all the instruments with a National Instrument card, and transferred all the data on a relatively modest PC with solid state drive (Kingston V300), a Intel Core i7 processor (3.5 GHz) and 16 GB of RAM. Typical datasets for single animals imaged every 20 minutes for 48 hours are 25 GB per imaging channel. As we typically acquire data for 15-20 animals in two imaging channels, one microscopy session results in 0.75-1.0 Tb of imaging data.

Action:

- We have added a full description of our computer hardware in the ‘Microscopy imaging’ section of the online methods (lines 334-337).

b The authors imply that the relatively small walls of the microchamber do not perturb development, yet it is obvious that by the end of the acquisition, the worm fills most of the microchamber volume. I would like to be more convinced that the confinement does not alter the underlying growth: the authors show in Supplementary Figure 1 that a very slightly larger chamber slightly affects the growth curve, and imply this is due to the lack of food. This point should be addressed (i.e., add more food to definitively test this hypothesis), and a significantly larger microchamber used to demonstrate that the walls do not significantly affect growth.

We decided not to address the Reviewer’s question using significantly larger microchambers, since in our experience microfabricating and ‘debugging’ the required molds as well as gaining experience in how to fill them with bacteria and eggs has been a relatively time-consuming process each time we changed the dimensions of our chambers. Instead, we performed what we think is a better test: on our time-lapse microscope, we imaged animals growing on normal bacterial lawns on agar plates. Even though this comes at the expense that, since we cannot do automated imaging, we have more limited time resolution (approximately every 12 hours) and have no information on the timing of ecdysis, it has the great advantage that we can directly compare it to growth under standard laboratory conditions. We found that body length as a function of time was very similar for growth on plates and in microchambers, even for late L4 larvae.

Action:

- We have added to Supplementary Figure 1 data on larval growth, as measured by body length, of animals on standard agar plates.
- We have added a comment on the similarity between growth in microchambers and on plates to the main text (lines 90-92).
- We have added a description of our procedure for imaging growth of animals on plates in the caption of Supplementary Fig. 1.

c related to this point: larva dimensions may have diameter > 20 um, while chamber "height" is 20 um. Are larvae compressed and what effects does this have on development?

We examined this in a quantitative manner by measuring the distance between left-right pairs of seam cell, which because of the animal's orientation indicate the dimensions of the body along the vertical axis. Using this measure, we find that for animals from the L3 larval stage onwards, the vertical dimension of the body increase beyond 20µm. This indicates that the surrounding hydrogel deforms to accommodate the larger dimensions of the animal's body. To examine whether this in turn also deforms the animal's body, we compared the vertical (dorsal-ventral) and horizontal (left-right) dimensions, with the latter measured from the transmitted light images. We find that the horizontal dimension is on average 20% larger than the vertical one, indicating that compression of the larval body is relatively limited.

As a general remark regarding both points b and c of Reviewer #1, we would like to point out that the *C. elegans* post-embryonic lineage was obtained by Sulston and Horvitz using a protocol where larvae were compressed between a layer of agar and a glass coverslip, forming an environment similar to what we use here. In addition, none of the developmental processes we analyzed in our manuscript show strong deviations in timing or otherwise from what has been described, even at the latest stages of development.

Action:

- We have added a new panel to Supplementary Figure 1 that shows the vertical and horizontal dimensions of the animal's body as a function of time.
- We have added a reference to this result to the main text (lines 90-92).

d I assume the units of the y axis in Fig. 2d are in hours. Please make explicit in the figure.

Action:

- We have added the correct units to Fig. 2d.

e In figures 2e, 2f, it might be helpful to explicitly indicate 'animal 1', 'animal 2', etc. above each lineage.

Action:

- We have followed this useful suggestion from the Reviewer.

f What is the y axis scale in Figures 4d, e?

Action:

- We have added labels with the correct fractions to the y-axes of Fig. 4d and 4e.

g. I could not find temperature info for microchambers when imaging. Temperature affects developmental timing so this should be specified in methods if not already present.

We thank the Reviewer for pointing out this important oversight. While we did report the temperature we used for normal culturing of strains, we forgot to report the (much more crucial) temperature at which we imaged. We currently have no temperature control on the microscope and rely on the ambient temperature in the microscopy room, which we have measured to be 22°C. Even though that temperature is well within the temperature regime at which development is ‘normal’ and, in fact, some *C. elegans* labs culture animals at that temperature, the more standard temperature is 20°C and we are currently in the process of adding temperature control to our setup to bring it further in line with standard culturing conditions.

Action:

- We have added a sentence on the temperature during microscopy imaging to the ‘Microscopy imaging’ section of the online methods. (lines 343-344)

h. analysis of Netrin mutations on DTC D-V migration: the number of cells with phenotypes (10 + 11) is greater than the number examined. Do some of the DTCs show multiple phenotypes? If so, this should be clarified, as in lines 187-190

The Reviewer is correct: we compared failure to move from dorsal to ventral (observed in 10/18 distal tip cells) with failure to change direction in the antero-posterior direction (11/18 cells). Since we find that these phenotypes occur at least partially independent of another the total number of cells with phenotypes is indeed larger than the total number of cells.

Action:

- We have modified the text to clarify this. (lines 190-191)

Reviewer #2

Gritti et al reported a microscopy method for long-term fluorescence time-lapse imaging of C. elegans larvae. They showed that they could use this protocol to study seam cell division, distal tip cell migration and gene expression. This technique can be useful in several cases such as the long-range distal tip cell migration. However, the following limitations preclude its publication in a journal such as Nature Communication.

1. The protocol is lack of sufficient spatial resolution for scientific discoveries. The authors used an X40 objective for imaging analysis; however, under such a setup, very limited subcellular structure can be visible. It is nowadays essential to study cellular processes during C. elegans development at the molecular or subcellular resolution. X40 objective does not enable the imaging of the cytoskeleton or polarity molecules or others. Seeing the cell morphology (e.g. distal tip cell,) or nuclei (seam cells) or global change of fluorescence (gene expression) will not advance our understanding of cellular mechanisms of these processes. They authors should demonstrate that X100 objective or even higher (e.g., X150) can fit in their protocol.

We believe this remark of Reviewer #2 is based on the misunderstanding that the resolution is limited by the magnification of the objective, whereas it is limited by the numerical aperture. We use a high (1.3) N.A. 40x oil-immersion objective, which is close to the numerical aperture of typical 100x objectives (with N.A of 1.4-1.5). By using a camera with large field of view but small pixel size, we can still obtain images of single cells with subcellular resolution. As an example, in Fig. 4F we show images of seam cells where we can clearly distinguish the nucleus and the cell membrane and where the overall resolution is certainly high enough to resolve spatial distributions of polarity or cytoskeletal proteins.

In principle, it is possible to increase the N.A. and resolution even further by using a higher magnification objective. This comes at the expense of having to use smaller microchamber dimensions to fit the microchamber into the reduced field of view. For instance, we have used a 1.4 N.A. 60x objective to image development in 190x190x10 μ m microchambers. With these dimensions, the amount of bacteria in the microchamber can only sustain growth through L1 and L2 and leads to arrest in early-mid L3. However, even with these limitations this is still a far longer total imaging time than is possible with current alternative time-lapse imaging approaches. We currently use this approach to study gene expression and cell signaling in stochastic cell fate decisions in the L2 larval stage.

Action:

- We rewrote the manuscript to more strongly emphasize that the use of a high NA 40x objective results in high enough spatial resolution to visualize subcellular features. (lines 73-76 and 253-256)

2. Similarly, the method does not offer the sufficient temporal resolution for discoveries. They authors claimed that their imaging time interval is 20 min, which is far too slow to document enough details. For example, the post-embryonic Q cell or P cell or seam cell division can be completed within 20 minutes. The previous study of Q cell asymmetric cell division used 20 second time interval to follow the dynamics of centrosome and myosin during divisions and even used 5 sec time interval to visualize membrane dynamics during Q cell division, leading to an important discovery in asymmetric cell division (Ou et al. Science 2010). Moreover, the migrating cells always use less than 10 minutes to establish polarity and reorganize the actin cytoskeleton after division. Again, the protocol described in this manuscript will not be possibly to yield any information to address such problem.

Limiting the time interval to 20 mins is necessary to sustain development over all four larval stages and, as the examples in our manuscript clearly show, is sufficient to image a wide range of fundamental developmental processes. However, because our setup is optimized for speed it is in principle capable of much shorter time intervals: 10-20ms for a single image and <0.5s for a stack of images of two fluorescent channels. It is therefore possible to image developmental processes at significantly higher time resolution, albeit for a limited time as larval development will ultimately arrest due to phototoxicity. However, similar limits due to phototoxicity also hold for the microscopy approach in the article referenced by the Reviewer, where larvae are immobilized. The main advantage of larval immobilization for applications such as Q cell division is that, in contrast to our approach, it can naturally be combined with standard confocal microscopy for better optical sectioning. That said, our approach is specifically designed for studying developmental processes that unfold over timescales much longer than the Q cell division and for which the immobilization approach is inadequate.

3. The major issue of this manuscript is that the lack of the demonstration of the potential of yielding novel biological insights. Any fancy techniques will be quickly faded if there is no

such potential regardless what the technique relies on.

Although the protocol may not be very valuable for cell biology, it may attract developmental biologists if (1) long-term (>10 hr) imaging is required and (2) the temporal/spatial resolution is not necessary. To this end, the manuscript will be very interesting to the readership in journals such as a Tool in Development.

The primary aim of this manuscript was to present our microscopy approach and demonstrate its power in imaging a diverse range of developmental processes currently inaccessible to time-lapse microscopy. Given this emphasis on broad applicability, we therefore chose not to focus in depth on studying a single example. However, even in this broader context we have shown clear examples of how such an approach can be used to gain new biological insight, for instance to analyze lineage errors in mutants (Fig. 2f,g) or the connection between distal tip cell migration and the molting cycle (Fig. 3g). In fact, all three examples discussed in the manuscript have become starting points for more in-depth studies, in direct collaboration with *C. elegans* groups. In addition, in our group we are currently using this approach to study gene expression dynamics in stochastic cell fate decisions in *C. elegans*. For such stochastic processes, where animal-to-animal variability is extremely high and the knowledge of the full history of the cell fate decision is essential to understand the underlying mechanisms, the ability to perform long-term time-lapse microscopy is essential.

We also think our approach potentially has great value for cell biology. Apart from the fact that in principle our setup is capable of imaging at a temporal resolution necessary to capture fast processes such as a Q cell division, we think there are many applications in cell biology in developmental processes that unfold at time scale slow enough that they are inaccessible to immobilization-based techniques. For instance, distal tip cell migration would be a very interesting system to visualize dynamics of cytoskeletal and cell polarity proteins during the entire migration trajectory. In addition, we currently also use our approach to study cell signaling dynamics during cell fate decisions.

More in general, the ability to study developmental dynamics at the single-cell level in growing and feeding animals has clear potential for novel biological insight, as also recognized by Reviewer #3. This is also reflected in the significant attention from biology groups we have already received based on conference presentations and that has led to a range of collaborations on topics as diverse as developmental timing, cell lineage dynamics, variability of development in wild *C. elegans* isolates and host-pathogen interactions.

Finally, the writing can be improved to be more scholarly.

1. Line 28-29, "However, fluorescence time-lapse microscopy is currently rarely used to study C. elegans post embryonic development." "rarely" may not be appropriate because a simple PubMed search revealed more than 20 primary research papers using live cell imaging of worm larvae.

We feel justified in using the term 'rarely'. Although, as the Reviewer points out, articles that perform time-lapse microscopy to study post-embryonic development using immobilization or microfluidics do exist, they form a very small fraction of the total body of literature on post-embryonic development, particularly where it concerns developmental processes that last more than 1-5 hours. This is particularly striking compared to the study of early embryonic development in *C. elegans*, where time-lapse microscope has quickly become an essential tool. However, we did change the text to emphasize that long-term time-lapse microscopy in particular is used rarely.

Action:

- We changed the text to emphasize that long-term time-lapse microscopy is used rarely. (lines 28+272)

2. Line 150: "*Q neuroblast migration has been imaged using timelapse microscopy for short (~1h) periods in immobilized C. elegans larvae*". *Q cell migration can be imaged for 3-4 hours (Ou & Vale, Journal of Cell Biology, 2009) without affecting the migration speed and distance compared with the original Sulston and Horvitz work (Dev Biol 1997). Q cell was only imaged for a couple of hours because its migration is completed within 3-4 hours. There is no point to image further. Incorrect citation/interpretation of the previous literature will not certainly help the novelty of the current protocol.*

We are aware that processes like Q cell migration and seam cell division can be imaged for up to 3-4 hours and merely wanted to express the order of magnitude. However, in our own personal experience using immobilization to study cell divisions in the seam cell and vulva lineage, we often find that even though some cells divide for up to 3-4 hours, others often arrest or show perturbed dynamics.

Action:

- We changed the text from "short (~1hr)" to "short(<3-4hr)". (line 152)

3. *The authors can emphasize more that this protocol can be complementary to the existing protocols: Chai et al. reported high resolution imaging of L1 larvae whereas this protocol, as the author stated, will not work for the L1 larvae because the high power of lasers must be used. However, the protocol enables the long-term imaging of post-L1 processes that do not require high spatial and temporal resolutions.*

We do not agree that our approach does not work for L1 larvae. In fact, our manuscript shows two examples (seam cell division and molting cycle gene expression) where we image single-cell dynamics during the L1 larval stage. In addition, we believe that in terms of photodamage our wide-field illumination approach is similar to the combination of confocal imaging and immobilization the Reviewer refers to. As we discussed under points 1 and 3 of Reviewer #3, our approach is likely able to operate at spatial and temporal resolutions similar to what is possible using the protocol of Chai et al., based on immobilization. However, the advantage of immobilization for short-term imaging is that it can be easily combined with standard confocal microscopy. We have stated this now more clearly in the manuscript.

Action:

- We changed the text to reflect that immobilization-based techniques are more straightforward to combine with confocal microscopy (lines 253-256).

Reviewer #3

Gritti and coauthors describe a system for imaging the entirety of C. elegans postembryonic development in living worms at single-cell resolution. They demonstrate the value of the system by imaging and quantitative analysis of three processes that take too long to image continuously previously: distal tip cell migration, seam cell divisions, and gene expression that oscillates with molting cycles.

I view this as a technical tour de force that can have a strong practical impact on diverse areas of biology that depend on imaging a living organism. The writing is clear, and the data are beautiful and well presented both in images and videos and in quantitative analyses.

We are happy with the positive opinion of Reviewer #3, particularly that the Reviewer believes our approach will have a strong practical impact.

I consider this manuscript as a very strong candidate for publication in Nature Communications (or Nature Methods), with just one major concern: The exposure times used were so short, and the fluorescent markers presumably so highly expressed, that it will be difficult for others to make practical use of this method in many cases. This was no doubt needed because the worms were moving. For this reason, I urge the authors to attempt combining the new method with their methods in their reference 22, for flowing in drugs that can reversibly arrest movement, and show that this allows longer exposure times needed for more dimly fluorescent cells.

We wish to note that our approach does not rely on the use of exceptionally highly expressed GFP reporters. Even though we use short exposure times, the laser pulses are sufficiently bright that we have been able to image a wide range of integrated transcriptional or translation reporters beyond those presented in the manuscript. Most of these reporters are not bright enough to be seen by eye under a fluorescence dissection microscope. While we have not yet tested single-copy integrations or CRISPR knock-ins, we have successfully imaged lin-39::GFP (wgIs18) and lin-12::GFP (wgIs72) reporters that were constructed by microparticle bombardment and integrated as low-copy transgenes. We therefore believe that our approach is not significantly limited in the range of reporters that can be imaged compared to other wide-field microscopy approaches.

While in principle it is possible to combine our hydrogel microchambers with a microfluidic layer to flow in paralyzing drugs, the downside of this is that it will significantly add to the technical complexity of our setup, both in terms of setting up the experiment and in controlling it in an automated way. We believe that a potentially more convenient solution to imaging lowly-expressed reporters in our setup is the use of brighter fluorescent proteins, such as mNeonGreen or mKate2, something we are already experimenting with ourselves.

Minor concerns:

Lines 24-25: Other nematode species have been lineaged too.

Action:

- We have changed that sentence to “nematodes such as *Caenorhabditis elegans* are currently the only animals where...” (lines 23-24)

Line 61: There should be some explanation of how only 20um tall is sufficient, since an adult worm is much thicker.

We refer to our response to Point c of Reviewer #1.

Line 134: MBA48 sounds like it must be a strain name, not a mutant name.

Action:

- We have changed “mutant, MBA48” to “mutant strain, MBA48” (line 136)

Line 429: "center of the midbody" is redundant.

Action:

- We have changed ‘midbody’ to ‘body’ throughout the text. (lines 381 and 458)

Reviewer #4

The paper by Gritti et al describes a technique to perform long-term microscopy on developing and moving C. elegans. The paper provides three beautifully done examples of how this technique is used in developmental biology and fluorescence live microscopy.

We are happy to hear the Reviewer’s positive remarks regarding the quality of the data acquired with our approach.

While the work presented in the paper is executed extremely nicely, the technical advancement is not terribly major. The basic idea is to push the microscopy system to have a short exposure time such that the otherwise possibly blurred images would not be. The way the authors demonstrated is by using a strong excitation source, very sensitive camera, and very importantly, quite bright markers (as the authors remarked in the text, "sufficiently bright"). This is perhaps the key limitation of the technique - that the markers have to be sufficiently bright and that one has to have access to fairly expensive microscopy equipment.

We believe that it is a strength of our approach that we use relatively ‘simple’ technical solutions to achieve long-term time-lapse microscopy. However, we do think our work represents a significant technical advancement, in terms of 1) the use of polyacrylamide microchambers as a tool to study C. elegans development, 2) the design of a microscopy setup capable of the fast acquisition speeds, large field of view and high spatial resolution required to image moving animals and 3) the development of tools to perform image analysis and quantitative data analysis on the time-lapse images we collect.

As discussed also in our response to Reviewer #3’s comments, our setup is not limited to the imaging of extremely bright fluorescent reporters: we have so far been able to acquire excellent data on almost all reporter strains we have imaged, including those known to be present at relatively low copy number. Our comment on sufficiently bright reporters was made in the context of the imaging of seam cells on both sides of the animal. Here, scattering by the intervening tissue can degrade the fluorescence signal to the extent that seam cells on side of the animal’s body most distant to the objective were poorly visible. However, this is a general problem for wide-field microscopy imaging and not specific to our setup. We discussed this in more detail for the wrt-2 reporter in the section on molting cycle genes.

We agree with the Reviewer’s comment that the full design as we present it in our manuscript requires a significant investment to set up. However, we believe that a more pared down version of our approach could already be sufficient to perform long-term time-lapse microscopy, although likely in a more limited fashion. For instance, it might be possible to combine polyacrylamide microchambers with LED illumination. LEDs are capable of short exposure times and will in the coming years likely increase rapidly in brightness.

The examples that the authors used are fairly well known developmental phenomena. While the authors observed that there exists an appreciable variability in each of the examples, and that the numbers of animals from these experiments are impressive, it doesn't seem that there is much novel biological insights from these experiments.

We first like to emphasize that our observation of ‘appreciable variability’ in animals imaged in parallel during the same experiment is in itself already novel and interesting. As we also discussed in response to point 3 of Reviewer #2, the main focus of our paper was on demonstrating the ability to image a wide range of developmental processes that were so far inaccessible to time-lapse microscopy. In this context, we were already able to provide new biological insight in the examples we studied. We are currently making extensive use of our technique to perform in-depth study of phenomena such as stochastic cell fate decisions or molting cycle oscillations, for which the temporal information our approach yields is essential.

Taken together, the main complaint this review has is that there is not sufficient technical or scientific advancement, even though the work is very beautifully done and the paper is clearly and thoughtfully written.

To summarize, we believe our approach represents a major technical advance in terms of the use of microfabricated structures, rapid image acquisition of moving animals and quantitative data and image analysis. In addition, the resulting quantitative data led to novel insight in the dynamics of cell division, migration and gene expression and uncovered previously unappreciated amounts of variability in these processes.

Reviewer #1 (Remarks to the Author)

The authors have successfully addressed my concerns; I have no further comments.

Reviewer #3 (Remarks to the Author)

The revision has satisfactorily addressed my concerns. This manuscript reports a valuable step forward in imaging live *C. elegans* despite the limitations raised by other reviewers, and I look forward to seeing the manuscript published.